# Research Trends in C-Terminal Domain Nuclear Envelope Phosphatase 1

**DOI:** 10.3390/life13061338

**Published:** 2023-06-07

**Authors:** Harikrishna Reddy Rallabandi, Haewon Choi, Hyunseung Cha, Young Jun Kim

**Affiliations:** Department of Medicinal Bioscience and Nanotechnology Research Center, Konkuk University, Chungju 27478, Republic of Korea

**Keywords:** CTDNEP1, LIPIN, nuclear envelope, tumor suppressor, protein phosphatase

## Abstract

C-terminal domain nuclear envelope phosphatase 1 (CTDNEP1, formerly Dullard) is a member of the newly emerging protein phosphatases and has been recognized in neuronal cell tissues in amphibians. It contains the phosphatase domain in the C-terminal, and the sequences are conserved in various taxa of organisms. CTDNEP1 has several roles in novel biological activities such as neural tube development in embryos, nuclear membrane biogenesis, regulation of bone morphogenetic protein signaling, and suppression of aggressive medulloblastoma. The three-dimensional structure of CTDNEP1 and the detailed action mechanisms of CTDNEP1’s functions have yet to be determined for several reasons. Therefore, CTDNEP1 is a protein phosphatase of interest due to recent exciting and essential works. In this short review, we summarize the presented biological roles, possible substrates, interacting proteins, and research prospects of CTDNEP1.

## 1. Introduction

Protein phosphatases are responsible for dephosphorylating phosphorylated amino acid residues of their substrate proteins [1]. Among protein phosphatases, a family of C-terminal domain (CTD) phosphatases (CTDPs) has recently been investigated in various systems [2]. The name CTD is taken from the C-terminal domain (CTD) of RNA polymerase II (RNAPII) because CTD phosphatase 1 (CTDP1) can dephosphorylate the phosphorylated residues of the CTD of RNAPII [3]. There are seven CTDPs in the human genome, which consist of CTD phosphatase 1 (CTDP1) [4], CTD small phosphatase 1 (CTDSP1) [5], CTD small phosphatase 2 (CTDSP2) [6], CTD small phosphatase-like (CTDSPL) [7], CTD nuclear envelope phosphatase 1 (CTDNEP1) [8], CTD small phosphatase-like 2 (CTDSPL2) [9], and ubiquitin-like domain-containing CTD phosphatase 1 (UBLCP1) [10]. CTDP1’s dephosphorylation results in the termination of RNAPII transcription and is necessary for cell survival [2]. CTDNEP1 is a member of the newly emerging CTD phosphatase family, and its various roles are presented in some studies. The presented biological roles of CTDNEP1 are summarized in Table 1.

CTDNEP1 was initially detected in neuronal tissues of *Xenopus laevis* [11]. The biochemical properties of CTDNEP1 have been characterized in a few studies [11,12], except for the three-dimensional structure. The expression of CTDNEP1 is shown in novel biological activities such as neural tube development in *Xenopus* embryos [11], nuclear envelope membrane biogenesis [12], and the critical regulation of bone morphogenetic protein (BMP) receptors [13]. According to a recent study [8], the loss of CTDNEP1 induces aggressive medulloblastoma.

Among the several roles of CTDNEP1, the mechanism of nuclear membrane biogenesis has been elucidated in relatively more detail. In yeast, the homolog protein of CTDNEP1, nuclear envelope morphology 1 (NEM1), dephosphorylates a phosphatic acid (PA) phosphatase, PA phosphohydrolase 1 (PAH1) [14,15,16]. Similarly, the human CTDNEP1 dephosphorylates a human PA phosphatase, LIPIN1, in the presence of insulin [12]. The novel property of CTDNEP1 in its substrate specificity for the dephosphorylation of serine/threonine residues in human LIPIN1 involves nuclear envelope formation [12]. The dephosphorylation action of CTDNEP1 on LIPIN1 results in nuclear membrane dynamics through several mechanisms. However, more studies are necessary to elucidate the downstream signaling mechanism after the dephosphorylation of PA.

BMP signaling is the fundamental biological process during embryogenesis [17], involving different developmental processes such as neuronal gene induction and neuronal pattering in amphibians [11,13]. CTDNEP1 regulates BMP signaling in various biological processes [17], such as the positive regulatory effect observed in wingless and int-1 (WNT) signaling in mice [18], the adverse regulatory effects in wing vein formation in fruit flies [19], and its role in kidney development and maintenance after birth [20]. WNT/β-catenin signaling is a cell–cell signaling mechanism [17]. It plays a crucial role during embryogenesis, mainly in cell proliferation and migration [21]. Loss of the protein phosphatase function of CTDNEP1 has shown downregulation of WNT signaling and a reduction in Dishevelled 2 (Dvl2), which fails to form primordial germ cells (PGCs) in mouse embryos [18]. This denotes that CTDNEP1 acts with neuronal cells and is associated with other biological functions. However, more detailed studies are needed to understand the mechanisms of CTDNEP1’s action on BMP and WNT/β-catenin signaling.

A recent study showed that loss of CTDNEP1 can induce aggressive brain tumors [8]. Mutations in CTDNEP1 have been discovered in MYC-driven medulloblastomas. Deficiency in CTDNEP1 activates MYC activity by elevating MYC phosphorylation, and triggers chromosomal instability. CTDNEP1 also modulates the activities of topoisomerase TOP2A and checkpoint kinase CHEK1 [8]. This work presents a novel and exciting role of CTDNEP1 as a tumor suppressor. However, detailed studies are necessary to understand how CTDNEP1 is involved in this biological process. The in-depth focus on this CTDNEP1 could present different dimensions of its activity. Therefore, this review summarizes the biological roles of CTDNEP1 and the research trends in CTDNEP1.

**Table 1 life-13-01338-t001:** Biological roles of CTDNEP1.

Roles	Possible Substrates/Interacting Proteins	References
Nuclear membrane biogenesis	LIPIN1 PAH1 SUN2	[22] [14] [23]
Nuclear pore complex insertion	Torsin	[24]
Nuclear positioning	Eps8L2	[25]
Neural induction	BMPR	[11,13]
Kidney formation	BMPR SMAD 1/5	[20,26] [27,28]
Primordial germ cell activation	unknown	[18]
Wing vein formation	pMAD DSRF	[19] [29,30]
Bone homeostasis Hemorrhage in the adult ovarian follicles	SMAD 2/3 unknown	[31] [32]
Tumor suppressor	MYC TOP2A CHEK1	[8] [8] [8]
Control of birthweight Hepatosteatosis	unknown unknown	[33] [34]

## 2. Biochemical Characterization of CTDNEP1

CTDNEP1, as a haloacid dehalogenase (HAD) superfamily phosphatase, has been identified in neural tissues during the screening of cells via whole-mount in situ hybridization in the neural region of *Xenopus* embryos [11]. The temporal and spatial distributions of CTDNEP1 were observed through whole-mount in situ hybridization [11]. Before gastrulation, CTDNEP1 is expressed in the animal hemisphere. In the early and mid-gastrula stages, expression is limited to the neural region. In the post-gastrula stages, it is distributed among neural tissues, branchial neural tissues, branchial arches, and pronephroi [11]. CTDNEP1 is shown to be a potential regulator of neural tube development in *Xenopus* [11] and localized to the nuclear envelope [12].

CTDNEP1 has been identified with 732bp and an amino acid sequence of 244 residues. In silico sequence alignment revealed that CTDNEP1 has a homolog of yeasts and humans [11]. CTDNEP1 contains a nuclear LIM interactor–interacting factor domain—in other words, the CTD phosphatase domain—in the C-terminal, and it is conserved in various taxa of organisms [11]. The sequence alignment of its homologs is shown in Figure 1, which shows that the sequences of CTDNEP1 homologs are highly conserved.

CTDNEP1 evolved from yeasts to humans and is expressed in both [12]. CTDNEP1 has transmembrane regions in its N-terminal end, and the membrane-spanning region is vital for its nuclear membrane localization [12]. Separate analyses and experimental procedures were performed to identify similarities between NEM1 and CTDNEP1, such as its localization priorities, substrate specificity, phosphorylation activity, and functional properties [12]. Structural studies revealed that N-terminal residues (1–45) are required for its localization into the nuclear envelope, and its C-terminal region has a conserved active site motif DXDX(T/V). Mutation studies in the CTDNEP1 active site motif region revealed the role of CTDNEP1. The conserved catalytic domain, CTD phosphatase domain, in its C-terminal region is a commonly shared motif among all organisms where the protein is present [12].

A phosphopeptide array assay observed that CTDNEP1 preferentially hydrolyzed pSer compared to pThr and pTyr [12]. This denotes that Ser106, the insulin-stimulated phosphorylation site for LIPIN1, could be a dephosphorylation site of CTDNEP1. A steady-state kinetic assay was used to determine the specificity constants of CTDNEP1 from LIPIN1-derived pSer residues and the related dephosphorylation mechanism [22]. Several nano-peptides from human LIPIN1 were considered substrates for CTDNEP1. The results showed that the highest k_cat_/K_m_ value was attained by the peptide Ser106, which is the insulin-dependent phosphorylation site in LIPIN1 [22]. It was also observed that CTDNEP1 preferred the insulin-dependent peptide Ser106, with a k_cat_/K_m_ value approximately 40 times higher than the next most reactive peptide, Ser248 [22]. Finally, from the observations, it is concluded that CTDNEP1 prefers the insulin-dependent phosphorylation site in LIPIN1, which is the key regulator of nuclear membrane biogenesis.

Several three-dimensional structures of CTD phosphatases, such as CTDSP1, CTDSP2, and CTDSPL, are available at https://www.rcsb.org/ (accessed on 2 April 2023). The model structure of CTDNEP1 was predicted using alphafold2 and is available at https://alphafold.ebi.ac.uk/search/text/CTDNEP1 (accessed on 2 April 2023). The prediction of the catalytic domain (46–244) of human CTDNEP1 shows high per-residue confidence scores, while the prediction of the N-terminal region (1–45) shows low per-residue confidence scores. The model structure of CTDNEP1 is compared with other CTD phosphatases’ structures in Figure 2, which shows the similarity of the catalytic domain, the CTD phosphatase domain of CTDNEP1, CTDSP1, CTDSP2, and CTDSPL. However, the three-dimensional structure of CTDNEP1 has yet to be experimentally determined to elucidate its diverse roles.

## 3. CTDNEP1 in Nuclear Membrane Biogenesis

A genetic interaction study in yeast identified the protein complexes NEM1 and sporulation 7 (SPO7). It plays a vital role in nuclear envelope morphogenesis by regulating PAH1 [12]. In one study, human CTDNEP1 was identified as a homolog to yeast NEM, a member of the nuclear envelope biogenesis cascade. Human nuclear envelope phosphatase 1-regulatory subunit 1 (NEP1-R1, formerly TMEM188) is the SPO7 ortholog and plays a role in nuclear envelope biogenesis [35]. The signaling pathway relating to CTDNEP1 is conserved from yeast to humans [12]. The loss of CTDNEP1 function results in aberration in nuclear membrane biogenesis. Active CTDNEP1 could rescue the phenotypic form of the nuclear envelope in yeast cells, but inactive CTDNEP1 could not [12]. In yeast, the NEM1-SPO7 complex dephosphorylates the yeast PA phosphatase PAH1 [14]. Similarly, CTDNEP1 was shown to dephosphorylate the mammalian ortholog LIPIN1 [22]. PAH1 and LIPIN1 have a similar catalytic motif, DIDGT [36], and both are members of the nuclear membrane biogenesis cascade [12]. The whole mechanism denotes the role of phosphatase cascade signaling in nuclear membrane biogenesis. This remarkable role of CTDNEP1 could be related to the effect of nuclear membrane biogenesis on brain development.

Human LIPIN1 is the ortholog to PAH1 in yeast. It acts as a PA phosphatase and is also one of the critical regulators of cellular lipid metabolism in nuclear membrane biogenesis and various tissues [37,38,39,40,41,42,43,44,45,46,47,48,49]. Polymorphism in LIPIN isoforms may lead to a poor condition. In yeast, NEM1 and SPO7 act on PAH1; conversely, in humans, CTDNEP1 and NEP1-R1 dephosphorylate LIPIN1. Therefore, the dephosphorylation of LIPIN1 by CTDNEP1 controls the de novo glycerolipid synthesis. CTDNEP1’s control of LIPIN1 can also regulate ER membrane biogenesis and result in the protection against chromosome mis-segregation [50]. Although detailed studies about how CTDNEP1 modulates LIPIN’s phosphorylation state to influence nuclear and ER membrane dynamics are necessary, active LIPIN1 produces diacylglycerol (DAG) from PA and results in the production of triglycerides (TAG) and the regulation of lipid homeostasis [51]. The regulation of lipid synthesis by CTDNEP1 controls nuclear envelope closure through the feeding and remodeling of ER membranes to nuclear envelope holes [52].

The dynamic interactions between nuclear components and inner nuclear membrane proteins regulate the nuclear architecture and functions. CTDNEP1 is related to a mechanism of regulated degradation of the SUN domain-containing protein 2 (SUN2) as one of the inner nuclear membrane proteins [23]. The accumulation of dephosphorylated SUN2 results in an abnormal nuclear architecture [23]. In addition, a recent study suggested that NEP1-R1 and CTDNEP1 phosphatase affect interphase nuclear pore complex insertion through lipid-dependent and lipid-independent mechanisms with Torsin ATPase [24]. CTDNEP1 controls nuclear positioning during cell migration by regulating actin cables using Eps8L2, a CTDNEP1-interacting partner in a yeast two-hybrid (Y2H) screening [25]. Therefore, several studies show that CTDNEP1 also has a role in adequately forming nuclear envelopes through different mechanisms, which should be studied in more detail.

## 4. CTDNEP1 in BMP-Mediated Biological Processes

BMP belongs to the transforming growth factor-β (TGF-β) superfamily involved in various biological activities. Its activity plays a vital role during embryogenesis [53]. Knockdown studies of CTDNEP1 showed the adverse effects of BMP signaling in different cancers and other diseases. A detailed study of BMP signaling elucidated the regulatory effects of CTDNEP1 in BMP signaling and its importance in various biological processes. CTDNEP1’s dephosphorylation activity has been determined to be vital in BMP-mediated biological processes such as neural induction [11,13], kidney formation [20], and wing vein formation [19].

The knockdown approach using an antisense morpholino oligonucleotide was initially employed in *Xenopus* embryos, and the expression of CTDNEP1 was reduced via translational silencing [11]. The effect of the impaired function of CTDNEP1 resulted in an incomplete closure of the notochord, deficiencies identified in neural development, poor eye formation, and disorganization of the midbrain, hindbrain, and spinal cord [11]. CTDNEP1 promotes the degradation of BMP receptors (BMPRs) via the lipid raft-caveolar pathway and represses the phosphorylation of BMPRs [13]. These actions on BMPRs of CTDNEP1 are essential for neural induction in *Xenopus* embryos [13]. Conversely, over-expression of CTDNEP1 was shown to have an adverse effect, leading to apoptosis, in *Xenopus* embryos [11]. These results show that CTDNEP1 is a critical factor in successfully proliferating the abovementioned biological processes. On the other hand, the responsible mechanism of CTDNEP1 in terms of these biological processes is still unclear.

BMP and its key regulators play a significant role in various stages of embryo development, and their role is verified in kidney development and maintenance after birth. Two BMP ligands (BMP4 and BMP7) and their receptors are mainly involved in various stages of kidney development [20]. BMP4 is expressed in the mesenchyme surrounding the nephric duct, and BMP7 is expressed in the metanephric mesenchyme and ureteric buds. BMP4 possesses different functions such as ureteric stalk elongation and regulation of budding, and branching of ureteric tips along with the BMP receptor type 1A (BMPR1A), also known as activin receptor-like kinase 3 (ALK3) [26]. In the mid-gestation stage, BMP7 and pro-BMP factor Crossveinless2 (Cv2), expressed in metanephric mesenchyme, collectively maintain the nephron progenitor cells. BMP7, along with BMP receptor type 1B (BMPR1B), also known as activin receptor-like kinase 6 (ALK6) and activin receptor-like kinase 2 (ALK2), activates the mitogen-activated protein kinase (MAPK) pathway and helps in the maintenance of nephron progenitor cells and ureteric buds. The BMP suppressor of mothers against decapentaplegic (SMAD) pathway helps in the proliferation of nephron progenitor cells [27].

CTDNEP1 plays an essential role in various embryonic stages by limiting BMP signaling and helps in kidney development and postnatal-stage kidney maintenance [20]. During embryonic kidney development, expression of CTDNEP1 was observed in the metanephric mesenchyme, ureteric buds, and collecting ducts [28]. CTDNEP1 modulates BMP signaling via its dephosphorylation activity and limits the phosphorylated SMAD 1/5 at a proper level. In CTDNEP1 mutant mice, an increased concentration of SMAD 1/5 was observed, lowering the primordial germ cell (PGC) number in the post-implantation stage [13]. CTDNEP1 dephosphorylates the BMP receptors (BMPR) and degrades the BMPR through caveolin-dependent ubiquitin-mediated proteasomal degradation to repress BMP signaling. Deletion of CTDNEP1 in postnatal kidneys affects nephron loss due to apoptosis and enlargement of the pelvic cavity because of an increase in SMAD 1/5 [20]. As CTDNEP1 acts at the receptor level and regulates the BMP activity, it is significant in kidney development and maintenance after birth [54].

Much evidence has proven that BMP signaling is a crucial element in embryonic development; in contrast, BMP plays an essential role in wing vein formation in *Drosophila Melanogaster* during the pupa stage [55]. In general, the *Drosophila* wing consists of five central longitudinal veins (LVs). Induction of CTDNEP1 showed measurable changes in BMP-mediated biological processes such as wing vein formation and wing disc formation. Lowering the concentrations of CTDNEP1 showed 38% different ectopic formations of cross veins (CVs) and LV broadening. The abnormality was observed due to the harmful modulation of BMP signaling in the pupation stage, which is the pre-adult form of the organism [19]. Negative modulation of BMP signaling by CTDNEP1 and its interaction with the components of the BMP signaling pathway were observed in genetic interaction assays. The effects of CTDNEP1 were measured using phosphorylated mothers against decapentaplegic (pMAD), which accumulates in the veins for BMP signaling [19]. In the hypomorphic mutant of CTDNEP1, ectopic accumulation of pMAD was observed where CV forms, and reduced activity of *Drosophila* serum response factor (DSRF) was shown, which resulted in different vein formations. Contrary to the experiment of the hypomorphic mutant, the elevated CTDNEP1 expression showed reduced pMAD and extended accumulation of DSRF levels, representing the negative effect of CTDNEP1 on BMP signaling. In addition to these sequential experiments, it has been proven that CTDNEP1 only affects the BMP signal in its downstream regions [29,30]. The abovementioned observations denote the inverse effects of CTDNEP1 against BMP signaling.

Independent research works have studied other biological roles of CTDNEP1. CTDNEP1 regulates bone homeostasis via suppression of TGF-β signaling [31], which is a similar mechanism to the repression of BMP signaling explained in the section above. Past research has used conditional CTDNEP1-deficient mice, which induced the upregulation of protein levels of phospho-SMAD 2/3. In addition, the same CTDNEP1 mutant mice showed hemorrhage in the adult ovarian follicles, which might be caused by increased BMP signaling [32]. Studies to uncover the mechanism of CTDNEP1 are essential for gaining a clear understanding of the abovementioned biological phenomena.

## 5. CTDNEP1 in Other Biological Processes

A brain tumor grows aggressively and shows biological heterogeneity. CTDNEP1 has been found in an integrative deep sequencing analysis of brain tumor samples [56]. A recent study found that MYC-driven medulloblastomas have the most significantly enriched mutation in CTDNEP1 [8]. Based on these observations, the authors tried to reveal the detailed mechanism of CTDNEP1 as a tumor suppressor in aggressive MYC-driven medulloblastomas. The deficiency in CTDNEP1 increased the level of phosphorylated MYC, topoisomerase TOP2A, and checkpoint kinase CHEK1. This work shows that CTDNEP1 could dephosphorylate the phosphorylated Ser62 of MYC, and TOP2A and CHEK1. Therefore, CTDNEP1 suppresses brain tumors by triggering MYC amplification and genomic instability. There are several articles showing the possibility of using CTD phosphatases as tumor suppressors [5,7,57,58,59,60,61,62,63]. CTDSP1 has also been reported as a phosphatase of MYC Ser62 in liver cancers [59]. Both CTDSP1 and CTDNEP1 seem to have similar activity against the phosphorylated Ser62 of MYC. A detailed study is necessary to disclose the difference between the action mechanism of CTDSP1 and CTDNEP1 on phosphorylated MYC. In addition, an investigation into the action of CTDNEP1 as a tumor suppressor is necessary to distinguish it from other protein phosphatases’ action on tumors.

A placental multi-omics study showed that CTDNEP1 is a candidate functional gene for birthweight [33]. It was found that DNA methylation influences the expression of CTDNEP1 in the placenta, and CTDNEP1 was identified through multi-trait colocalization experiments. Expression of miR-122 is specific to the liver and involved in lipid metabolism (hepatosteatosis) [34]. A real-time PCR study and Western blot suggested that miR-122 represses CTDNEP1 and LIPIN1 in the triacylglycerol (TAG) pathway. These results only show the possibility of CTDNEP1 having novel roles. Therefore, new studies are required to understand its mechanisms in more detail.

## 6. Discussion and Conclusions

CTDNEP1 has significant roles in various biologically essential processes, such as neural tube formation in *Xenopus laevis* [11], nuclear envelope formation [12], regulation of BMP signaling in various embryonic development processes [13,20,54], and suppression of brain tumors [8]. However, the detailed mechanisms of CTDNEP1 in various critical biological activities are still unclear. We present the challenging research topics of CTDNEP1 in Table 2.

Among the CTD phosphatase family, CTDNEP1 is a transmembrane protein with 244 amino acid sequences [12]. It is highly conserved from yeasts to humans, and its biochemical properties are well-studied in ref. [12], excluding the three-dimensional structure. We expect the three-dimensional structure of CTDNEP1 to eventually be presented through X-ray crystallography, NMR, or Cryo-EM. In addition, the complex structures of CTDNEP1 and its substrates still need to be determined in order to gain a clear understanding of its action mechanisms. Although the interaction between CTDNEP1 and NEP1-R1 is known, a detailed description of their interacting mechanism is also necessary. There is a need to identify a regulatory material of CTDNEP1 to extend the biochemical characteristics of CTDNEP1, even though miR-122 is known as a repressing material of CTDNEP1.

The mechanism of CTDNEP1 in nuclear envelope biogenesis is presented in several studies [12,35,51,52]. Human CTDNEP1 is the ortholog of yeast NEM1, which preferentially dephosphorylates the Ser residues at the 106th position on LIPIN1 in the insulin-mediated LIPIN activation pathway [22]. It is required for nuclear membrane biogenesis. In addition, CTDNEP1 is related to the regulation of SUN2 and Eps8L2, which results in the proper nuclear architecture and positioning [23,25]. CTDNEP1, NEP1-RE1, and Torsin also regulate the nuclear pore complex insertion [24]. A recent review summarizes the participation of CTDNEP1 in lipid synthesis and nuclear envelope remodeling [51]. However, it is expected that novel studies on the mechanism of CTDNEP1’s involvement in nuclear membrane formation and its effects will be presented in more detail. The upstream signaling of CTDNEP1’s action on LIPIN and downstream signaling after PA dephosphorylation by LIPIN are needed in order to determine the detailed mechanisms. We expect the generalization of nuclear envelope formation regulated by CTDNEP1 and NEP1-R1 through the identification of novel substrates and combinations with previous studies.

Among all the dephosphorylation activities of CTDNEP1, the negative regulation against BMP in various biological and development processes has been studied [13,17,55,64]. The regulatory effects of CTDNEP1 in kidney formation during birth and nephron maintenance in the postnatal stage have shown proper regulation of BMP signaling at appropriate levels. CTDNEP1 actively regulates BMP signaling by dephosphorylating its receptors involved in the signaling process. It was also shown how CTDNEP1 regulates BMP signaling by dephosphorylating the downstream effectors. This negative modulatory effect of CTDNEP1 against BMP was studied in the fruit fly. pMAD and DSRF accumulation represented the modulation of BMP signaling by CTDNEP1 in the differential development process in the fruit fly. The modulatory effects of CTDNEP1 on bone homeostasis and hemorrhage in adult ovarian follicles are also related to BMP signaling [32].

The identification of novel substrates of CTDNEP1 in a novel BMP-related signaling pathway is still required to extend the biological function of CTDNEP1. However, we have a question concerning the direct dephosphorylation of BMPR or SMAD by CTDNEP1, because the subcellular localization of CTDNEP1 is the nuclear envelope, which is different from the subcellular localization of BMPR and SMAD. If CTDNEP1 directly dephosphorylates BMPR or SMAD, it is necessary to determine how they meet in the cellular compartment. If not, an indirect pathway of dephosphorylation is expected to be present. The different aspects of CTDNEP1’s action on BMP signaling compared to other protein phosphatases’ action on BMP signaling should be investigated because several protein phosphatases are well-known for their action on BMP signaling. The effects of CTDNEP1 against BMP signaling are unclear in many conditions. In the study of wing vein formation, there is no clear evidence of how CTDNEP1 acts on pMad at a molecular level. Hence, it is vital to focus on the mechanism of CTDNEP1 in regulating BMP signaling activity because BMP has an active role in many developmental processes and organ maintenance in postnatal stages.

In a recent study, CTDNEP1 was also found to be a tumor suppressor [8]. It was shown that the loss of CTDNEP1 increases the level of phosphorylated MYC, TOP2A, and CHEK1. However, it is still being determined whether CTDNEP1 directly dephosphorylates these proteins. A kinetic assay of CTDNEP1 on the proteins is necessary to understand its detailed mechanism as a tumor suppressor. The difference between CTDNEP1’s action and other protein phosphatases’ action on tumor signaling should be studied because several protein phosphatases are well-known as tumor suppressors. The identification of a novel tumor regulated by CTDNEP1 will still be of interest in extending our understanding of the biological roles of CTDNEP1. Targeting mutant CTDNEP1 tumors should be investigated to find novel therapeutic strategies for cancer, which has previously been attempted with mutant p53 tumors [65,66]. A few studies using multi-omics and micro-RNA show the possibility of controlling birthweight and hepatosteatosis [33,34]. Detailed research on these is necessary to reveal novel biological roles of CTDNEP1. There is a need to investigate the effects of micro-RNA on other CTDNEP1-related signaling pathways.

In summary, determining the three-dimensional structure of CTDNEP1 and the detailed action mechanisms of cellular processes regulated by CTDNEP1 is necessary to understand the biological functions and related diseases of the CTD phosphatase family. We assume that novel suggestions on the new biological roles and detailed mechanisms of CTDNEP1 will eventually be presented in studies using diverse systems and regulatory materials.

## Figures and Tables

**Figure 1 life-13-01338-f001:**
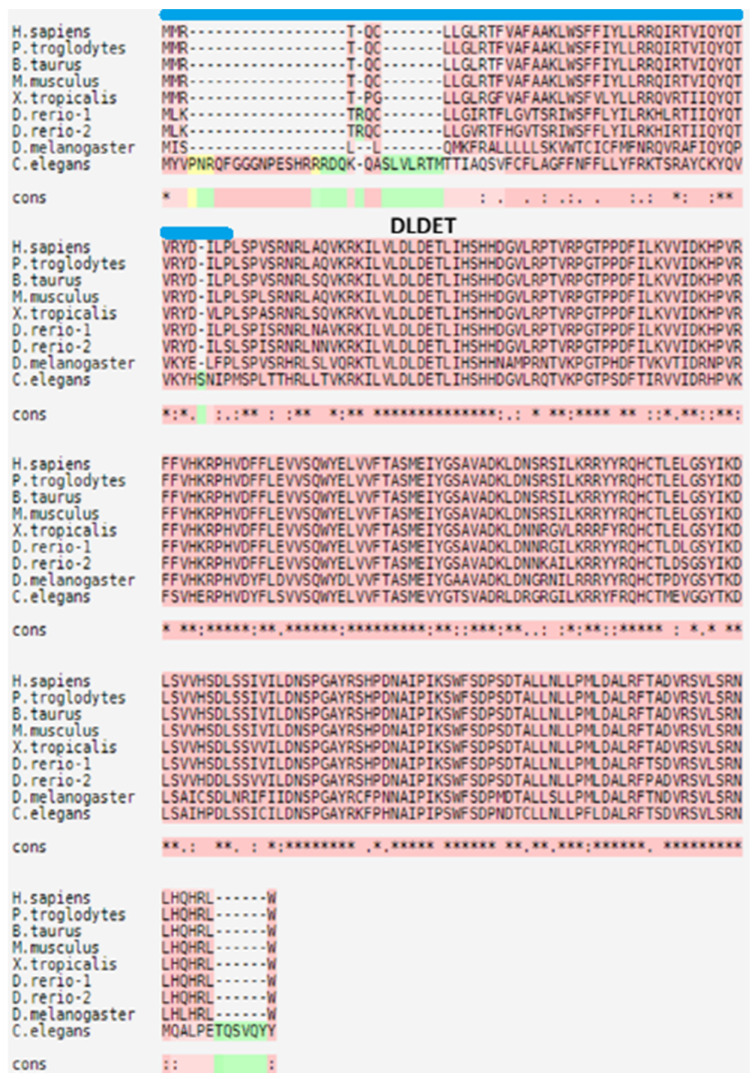
The multiple sequence alignments of CTDNEP1 homologs produced using T-Coffee (https://tcoffee.crg.eu/, accessed on 19 March 2023). GenBank accession numbers corresponding to the homologs are given below; Homo sapiens, NP_0001137247.1; Pan troglodytes, XP_511976.3; Bos taurus, NP_001039491.1; Mus musculus, NP_080293.1; Xenopus tropicalis, NP_001017177.1; Danio rerio, NP_001007310.1, NP_001007441.1; Drosophila melanogaster, NP_608449.1; Caenorhabditis elegans, NP_001254123.1. DLDET represents the active site sequences, and the transmembrane region is presented as a blue bar. The dark pink colors are reliable alignments, while the green colors are unreliable alignments. The yellow colors are moderately reliable alignments. The cons means the degree of conservation observed in each column, which is denoted by the following symbols; an * (asterisk) indicates positions which have a single, fully conserved residue; a : (colon) indicates conservation between groups of strongly similar properties; a . (period) indicates conservation between groups of weakly similar properties.

**Figure 2 life-13-01338-f002:**
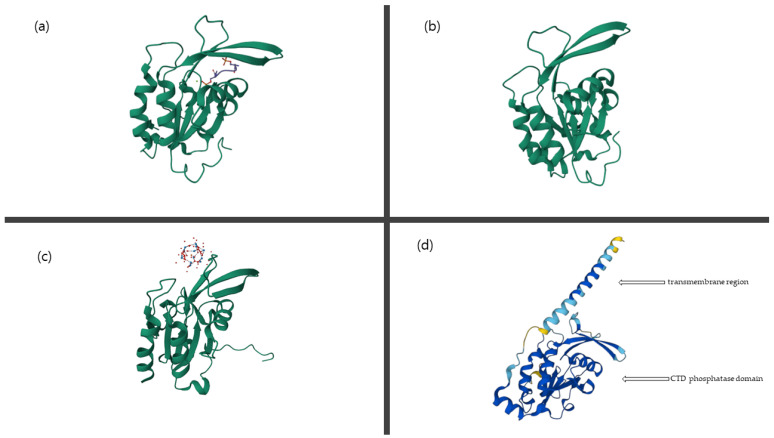
The structural comparison of human CTDNEP1 (**d**), a model structure predicted using alphafold2) with human CTDSP1 (**a**), pdb ID = 2ghq, a complex structure of the green-colored CTD phosphatase domain of CTDSP1 with the other colored CTD peptide of RNAPII), human CTDSP2 (**b**), pdb ID = 2q5e, an apo structure of the green-colored CTD phosphatase domain of CTDSP2), and human CTDSPL (**c**), pdb ID = 2hhl, a complex structure of the green-colored CTD phosphatase domain of CTDSPL with the other colored phosphatase inhibitor). In the model structure of CTDNEP1, the blue color represents very high per-residue confidence (pLDDT > 90), and the sky-blue color represents high confidence (90 > pLDDT > 70). The yellow color represents low confidence (70 > pLDDT > 50).

**Table 2 life-13-01338-t002:** Challenging research topics of CTDNEP1.

Research Fields	Research Topics
Biochemical characterization	Determination of the three-dimensional structure of CTDNEP1. Identification of interacting mechanism between CTDNEP1 and NEP1-R1. Determination of complex structure of CTDNEP1 and its substrates (e.g., LIPIN1). Kinetic investigation of CTDNEP1 with its possible substrates or interacting proteins. Identification of a regulatory material of CTDNEP1.
Nuclear envelope formation	Determination of upstream signaling of CTDNEP1. Identification of downstream signaling after PA dephosphorylation. Identification of a novel mechanism for nuclear membrane biogenesis.
BMP signaling	Identification of novel substrates of CTDNEP1 as a BMP signaling regulator. Determination of whether CTDNEP1 directly dephosphorylates BMPR, SMAD, or pMAD. If not, the identification of the indirect pathway. Identification of a novel BMP-related signaling. Characterization of the different aspects of CTDNEP1 from other protein phosphatases’ action on BMP signaling.
Tumor suppressor	Identification of novel substrates of CTDNEP1 as a tumor suppressor. Characterization of the different aspects of CTDNEP1 from other protein phosphatases’ action on tumor signaling. Identification of a novel related tumor. Investigation on targeting mutant CTDNEP1 tumors.
Another signaling	Determination of the mechanism of CTDNEP1 in birthweight control and hepatosteatosis. Investigation of miR-122 effect on CTDNEP1-related signaling.

## Data Availability

No new data were created.

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
