# Peer review of "Research Trends in C-Terminal Domain Nuclear Envelope Phosphatase 1"

_life, 2023, doi:10.3390/life13061338_

Round 1
Reviewer 1 Report
Protein phosphorylation represents one of the most common and versatile post-translational modification (PTM) allowing to the cells to respond to a wide array of clues. As most of the cellular PTMs, phosphorylation is a reversible process, in which kinases and phosphatases are counterbalancing each other. While kinases transfer, and covalently conjugate, a phosphate group to an amino-acid (Ser, Thr or Tyr) phosphatases remove it. The C-terminal domain nuclear envelope phosphatase 1 (CTDNEP1) has recently emerged as pivotal player controlling several biological processes ranging from embryonic neural tube development, to nuclear envelope biogenesis up to BMP receptor signal transduction. Eventually, its role it is not restricted only to physiological processes but it appears to be pivotal also in malignat neoplasms, including medulloblastoma. Structurally CTDNEP1 appears to be quite well conserved among different phyla. Nonetheless, though for most of the C terminal domain phosphatases (CTDPs) family members a 3D structure is present, to date a CTDNEP1 3D structure is still an unmet need. In their manuscript review titled "Research Trend on C-terminal Domain Nuclear Envelope Phosphatase" the authors summarize the current knowledge on the topic and discuss the future challenges. Overall, the manuscript flows quite smoothly. However, prior to publication it requires few amendments that are shortly below listed. The paragraphs should be carefully revised and optimized. I noticed that many biochemical properties are missing from the paragraph 2, titled " Biochemical Characterization of CTDNEP1". Conversely, many of them including structural domains, enzymatic features, substrate specificity preference, etc... are present in the paragraph titled " CTDNEP1 in Nuclear Membrane Biogenesis". This is a little bit confusing. Furthermore, the authors describe the transmembrane domains as present in the amino-terminal region of the protein. If one look at the paper published in JBC Volume 287, Issue 5, January 2012, Pages 3123-3137 it does not look so. Transmembrane regions are encompassed in the first half of the protein. Hence, I would recommend to revise carefully the paragraphs and update figure 1 detailing the transmembrane regions. Additionally, to the reviewer there are a couple of things, related to Figure 1, that are not clear: a) what is the meaning of the first block of the alignment in which there are no sequences but just figures? If there is a meaning, please this should explained, otherwise remove that part because seems meaningless; b) the color code require to be detailed. When it comes to Figure 2 the color code has to be detailed too, and the enzymatic domain highlighted/indicated. Ultimately, I noticed quite a few repetitions, thus I advice that the manuscript would benefit from a trimming. A few typos are scattered throughout the main text.English language is fine and minor editing is required, mostly typos.
Author Response
Reviewer 1
Protein phosphorylation represents one of the most common and versatile post-translational modification (PTM) allowing to the cells to respond to a wide array of clues. As most of the cellular PTMs, phosphorylation is a reversible process, in which kinases and phosphatases are counterbalancing each other. While kinases transfer, and covalently conjugate, a phosphate group to an amino-acid (Ser, Thr or Tyr) phosphatases remove it. The C-terminal domain nuclear envelope phosphatase 1 (CTDNEP1) has recently emerged as pivotal player controlling several biological processes ranging from embryonic neural tube development to nuclear envelope biogenesis up to BMP receptor signal transduction. Eventually, its role it is not restricted only to physiological processes but it appears to be pivotal also in malignant neoplasms, including medulloblastoma. Structurally CTDNEP1 appears to be quite well conserved among different phyla. Nonetheless, though for most of the C terminal domain phosphatases (CTDPs) family members a 3D structure is present, to date a CTDNEP1 3D structure is still an unmet need. In their manuscript review titled "Research Trend on C-terminal Domain Nuclear Envelope Phosphatase" the authors summarize the current knowledge on the topic and discuss the future challenges. Overall, the manuscript flows quite smoothly.
- Thanks for your comments!
However, prior to publication it requires few amendments that are shortly below listed. The paragraphs should be carefully revised and optimized.
à We revised and optimized the manuscript as much as possible.
I noticed that many biochemical properties are missing from the paragraph 2, titled " Biochemical Characterization of CTDNEP1". Conversely, many of them including structural domains, enzymatic features, substrate specificity preference, etc... are present in the paragraph titled " CTDNEP1 in Nuclear Membrane Biogenesis". This is a little bit confusing.
- We moved some sentences to the section of Biochemical Characterization of CTDNEP1 and rewrite as a reviewer suggested, written in blue.
Furthermore, the authors describe the transmembrane domains as present in the amino-terminal region of the protein. If one look at the paper published in JBC Volume 287, Issue 5, January 2012, Pages 3123-3137 it does not look so. Transmembrane regions are encompassed in the first half of the protein.
- The transmembrane region, which is shown in JBC Volume 287, Issue 5, January 2012, Pages 3123-3137, is misrepresented. They tried to present the transmembrane region of nuclear envelope phosphatase 1-regulatory subunit 1 (NEP1-R1, formerly TMEM188), which has the transmembrane region in the first half of the protein. Therefore, they just showed the illustration of CTDNEP1 as an interacting protein in the nuclear envelope and ignored the proper orientation of CTDNEP1 in nuclear envelope. According to ref. 12, transmembrane region is N-terminal 1-45 of CTDNEP1 because the CTDNEP1 without 1-45 amino acids is not localized into the nuclear envelope.
Hence, I would recommend to revise carefully the paragraphs and update figure 1 detailing the transmembrane regions. Additionally, to the reviewer there are a couple of things, related to Figure 1, that are not clear: a) what is the meaning of the first block of the alignment in which there are no sequences but just figures? If there is a meaning, please this should explained, otherwise remove that part because seems meaningless; b) the color code require to be detailed.
- Thanks again for your suggestion. We added the blue bar in figure 1 detailing the transmembrane region. In addition, we removed the first block of the alignment and added the detailed comments for the color codes as a reviewer suggested.
When it comes to Figure 2 the color code has to be detailed too, and the enzymatic domain highlighted/indicated.
- We also added the detailed comments for the color codes and showed the instruction for the enzymatic domain and transmembrane region in figure 2.
Ultimately, I noticed quite a few repetitions, thus I advice that the manuscript would benefit from a trimming.
- We changed and removed several sentences to remove the repetitions, as many as possible.
A few typos are scattered throughout the main text.
- We corrected the typos.
Comments on the Quality of English Language : English language is fine and minor editing is required, mostly typos.
- We asked the English editing through MDPI service and attached the English-editing-certificate.

Reviewer 2 Report
Manuscript summarizes the biological roles of CTDNEP1 and the research trend on CTDNEP1.
The manuscript is divided in 7 sections:
1. Introduction
2. Biochemical characterization of CTDNEP1
3. CTDNEP1 in nuclear membrane biogenesis
4. CTDNEP1 in BMP mediated biological processes
5. CTDNEP1 in other biological processes
6. Discussion and conclusion (wrongly numbered as 5)
7. References
Authors provide general information related to C-terminal domain phosphatases, among those phosphatases, CTDNEP1 is included. The biological effects of CTDNEP1 are cited or mentioned and listed in the Table 1.
It is indicated that novel functions for CTDNEP1 have been described but the references are not recent, but from the years 2002 (ref. 16), 2007 (ref. 17), and 2006 (ref. 18).
Most of the information provided and related to biological effects of CTDNEP1 is very concise. An example of this is the following expression:
“A recent study shows that loss of CTDNEP1 can induce aggressive brain tumors [8]. This work presents a novel and exciting role of CTDNEP1 as a tumor suppressor. However, detailed studies are necessary to understand how CTDNEP1 is involved in these biological processes”. Therefore, it is necessary to provide more information about the cellular mechanisms that occur in order to have a better understanding of the role of CTDNEP1. Only the processes of genesis of nuclear membrane and kidney development are moderately described.
The “Discussion and Conclusion” section contains repetitive information previously provided in the text of the manuscript.
It is necessary to highlight the biological impact of the topic addressed as well as elaborate a more descriptive article.
Author Response
Reviewer 2
Manuscript summarizes the biological roles of CTDNEP1 and the research trend on CTDNEP1.
The manuscript is divided in 7 sections:
- Introduction
- Biochemical characterization of CTDNEP1
- CTDNEP1 in nuclear membrane biogenesis
- CTDNEP1 in BMP mediated biological processes
- CTDNEP1 in other biological processes
- Discussion and conclusion (wrongly numbered as 5)
- References
Authors provide general information related to C-terminal domain phosphatases, among those phosphatases, CTDNEP1 is included. The biological effects of CTDNEP1 are cited or mentioned and listed in the Table 1.
It is indicated that novel functions for CTDNEP1 have been described but the references are not recent, but from the years 2002 (ref. 16), 2007 (ref. 17), and 2006 (ref. 18).
- Thanks for your comments! We described more recent references as many as possible, which are published in 2019 (ref. 52), 2020 (ref. 16), 2021 (ref. 24, 25, 50, 51), 2022 (ref. 14, 15, 23, 33), and 2023 (ref. 8).
- However, the references from years 2002 (ref. 11), 2007 (ref. 12), and 2006 (ref. 13) are necessary for the description of this review because they are the starting articles of CTDNEP1.
Most of the information provided and related to biological effects of CTDNEP1 is very concise. An example of this is the following expression:
“A recent study shows that loss of CTDNEP1 can induce aggressive brain tumors [8]. This work presents a novel and exciting role of CTDNEP1 as a tumor suppressor. However, detailed studies are necessary to understand how CTDNEP1 is involved in these biological processes”. Therefore, it is necessary to provide more information about the cellular mechanisms that occur in order to have a better understanding of the role of CTDNEP1. Only the processes of genesis of nuclear membrane and kidney development are moderately described.
à Thanks for your comments! We described more information as a tumor suppressor as much as possible, written in blue. The study of CTDNEP1 as a tumor suppressor is very recent, so there is too few studies to describe in more detail. Instead of it, we added information of the other phosphatases as tumor suppressors.
The “Discussion and Conclusion” section contains repetitive information previously provided in the text of the manuscript.
à We changed and removed several sentences to remove the repetitions as many as possible in the ‘Discussion and Conclusion section’, written in blue.
It is necessary to highlight the biological impact of the topic addressed as well as elaborate a more descriptive article.
à We changed and added several sentences to address the research topics to highlight the biological impact and to make this review more descriptive.

Reviewer 3 Report
Harikrishna Reddy Rallabandi and colleagues present a quality and well-written review manuscript describing the research trend on C-terminal domain nuclear envelope phosphatase 1.
Authors summarize the biological roles, possible substrates, interacting proteins, and research prospects of CTDNEP1.
In this work authors present a novel and exciting role of CTDNEP1 as a tumor suppressor. However, detailed studies are necessary to understand how CTDNEP1 is involved in these biological processes. The in-depth focus on this CTDNEP1 could present different dimensions of its activity. Therefore, this review summarizes the biological roles of CTDNEP1 and the research trend on CTDNEP1.
Finally, authors conclude that determining the three-dimensional structure of CTDNEP1 and the detailed action mechanism of cellular processes regulated by CTDNEP1 is necessary to understand the biological functions and related diseases of the CTD phosphatase family. They present the challenging research topics of CTDNEP1 and assume a novel suggestion on new biological roles and detailed mechanisms of CTDNEP1 from studies using diverse systems sooner or later.
Overall, the manuscript is highly valuable for the scientific community and should be accepted for publication after the corrections are made.
=====================
Other comments:
1) Please check for typos throughout the manuscript.
2) Authors are kindly encouraged to cite the following article that describes targeting mutant tumor suppressor using novel therapeutic approach that in principle can be useful for CTDNEP1 variants. DOI: 10.3389/fonc.2020.01460
Author Response
Reviewer 3
Harikrishna Reddy Rallabandi and colleagues present a quality and well-written review manuscript describing the research trend on C-terminal domain nuclear envelope phosphatase 1. Authors summarize the biological roles, possible substrates, interacting proteins, and research prospects of CTDNEP1. In this work authors present a novel and exciting role of CTDNEP1 as a tumor suppressor. However, detailed studies are necessary to understand how CTDNEP1 is involved in these biological processes. The in-depth focus on this CTDNEP1 could present different dimensions of its activity. Therefore, this review summarizes the biological roles of CTDNEP1 and the research trend on CTDNEP1. Finally, authors conclude that determining the three-dimensional structure of CTDNEP1 and the detailed action mechanism of cellular processes regulated by CTDNEP1 is necessary to understand the biological functions and related diseases of the CTD phosphatase family. They present the challenging research topics of CTDNEP1 and assume a novel suggestion on new biological roles and detailed mechanisms of CTDNEP1 from studies using diverse systems sooner or later. Overall, the manuscript is highly valuable for the scientific community and should be accepted for publication after the corrections are made.
- Thanks for your comments!
Other comments:
- Please check for typos throughout the manuscript.
à We corrected the typos as many as possible.
- Authors are kindly encouraged to cite the following article that describes targeting mutant tumor suppressor using novel therapeutic approach that in principle can be useful for CTDNEP1 variants. DOI: 10.3389/fonc.2020.01460
We cited two references for targeting mutant CTDNEP1 as a reviewer suggested.

Round 2
Reviewer 2 Report
The authors performed the changes requested by this reviewer.